# Gut Microbiota as Emerging Players in the Development of Alcohol-Related Liver Disease

**DOI:** 10.3390/biomedicines13010074

**Published:** 2024-12-31

**Authors:** Wei Li, Wenkang Gao, Shengqi Yan, Ling Yang, Qingjing Zhu, Huikuan Chu

**Affiliations:** 1Wuhan Jinyintan Hospital, Tongji Medical College of Huazhong University of Science and Technology, Hubei Clinical Research Center for Infectious Diseases, Wuhan Research Center for Communicable Disease Diagnosis and Treatment, Chinese Academy of Medical Sciences, Joint Laboratory of Infectious Diseases and Health, Wuhan Institute of Virology and Wuhan Jinyintan Hospital, Chinese Academy of Sciences, Wuhan 430023, China; 15913162529@163.com; 2Division of Gastroenterology, Union Hospital, Tongji Medical College, Huazhong University of Science and Technology, Wuhan 430022, China; gwkmed@163.com (W.G.); yansq1997@163.com (S.Y.); yanglinguh@hust.edu.cn (L.Y.)

**Keywords:** alcohol-related liver disease, gut microbiota, gut microbiota metabolites, intestinal barrier, gut–liver axis, inflammatory response

## Abstract

The global incidence and mortality rates of alcohol-related liver disease are on the rise, reflecting a growing health concern worldwide. Alcohol-related liver disease develops due to a complex interplay of multiple reasons, including oxidative stress generated during the metabolism of ethanol, immune response activated by immunogenic substances, and subsequent inflammatory processes. Recent research highlights the gut microbiota’s significant role in the progression of alcohol-related liver disease. In patients with alcohol-related liver disease, the relative abundance of pathogenic bacteria, including *Enterococcus faecalis*, increases and is positively correlated with the level of severity exhibited by alcohol-related liver disease. Supplement probiotics like *Lactobacillus*, as well as *Bifidobacterium*, have been found to alleviate alcohol-related liver disease. The gut microbiota is speculated to trigger specific signaling pathways, influence metabolite profiles, and modulate immune responses in the gut and liver. This research aimed to investigate the role of gut microorganisms in the onset and advancement of alcohol-related liver disease, as well as to uncover the underlying mechanisms by which the gut microbiota may contribute to its development. This review outlines current treatments for reversing gut dysbiosis, including probiotics, fecal microbiota transplantation, and targeted phage therapy. Particularly, targeted therapy will be a vital aspect of future alcohol-related liver disease treatment. It is to be hoped that this article will prove beneficial for the treatment of alcohol-related liver disease.

## 1. Introduction

Alcohol-related liver disease (ALD) includes various conditions, comprising asymptomatic hepatic steatosis, alcohol-induced steatohepatitis, fibrosis, cirrhosis, and hepatocellular carcinoma (HCC) [1,2,3] (Figure 1). A Markov modeling study predicted that from 2022 to 2040, without intervention, the age-standardized incidence of alcoholic cirrhosis in China is anticipated to increase from 10.66 to 26.27 per 100,000 person-years, indicating an estimated rise of approximately 146.5% [4]. In 2015, approximately 160,000 individuals in mainland China succumbed to alcohol-related deaths [5]. The presence of acute alcoholic hepatitis (AH) and liver cirrhosis is indicative of a high mortality rate, with patients diagnosed with advanced cirrhosis having a median survival of 1–2 years [6]. Within American borders, the national age-standardized mortality rate (AAMR) for ALD demonstrated a substantial increase of 23.4% between the years 2019 and 2020 [7]. Therefore, ALD has brought a heavy economic burden to society.

The pathogenesis of ALD arises from the intricate interplay of multiple aspects, including genetics and epigenetics influences, oxidative stress, an inflammatory response, hepatocellular damage and death, liver fibrosis, and a disruption in the gut microbes balance. A considerable amount of study has established an alteration in the intestinal microbes species and related metabolic pathways in ALD, offering a deeper understanding of this disease and potential avenues for its management [8,9,10,11,12,13]. Alcohol intake caused a marked diminution in gut microbes’ diversity. What is more, the population of beneficial bacteria declined while the prevalence of pathogenic bacteria expanded [14]. Research indicates that differences in gut microbiota composition significantly influence mice’s susceptibility to ALD [15]. Microbiota with a higher tendency to induce ALD are linked to a greater likelihood of severe liver injury and an intensified inflammatory response [16]. These results indicate that modulating the gut microbes may present a promising strategy for managing ALD. Therefore, it is important to conduct further research into the specific effects of gut microbiota on ALD and its associated mechanisms, as well as to assess the effectiveness and safety of various potential treatment options, including probiotic supplementation, fecal microbiota transplantation (FMT), and phage-based therapies. This review examines alterations in gut microbiota in ALD patients, highlighting how intestinal dysbiosis exacerbates ALD progression by triggering imbalanced immune responses in the intestine and hepatic tissue, leading to chronic inflammatory responses and metabolic disorders. Furthermore, it seeks to offer new perspectives on treating ALD by manipulating the microbiota.

## 2. Gut–Liver Axis

The gut–liver axis plays a vital role in ALD development, involving key components such as the gut, portal vein, and liver [17] (Figure 2). The portal vein functions as an essential pathway, facilitating communication between the gastrointestinal tract and the liver. This barrier is mainly comprised of a superficial mucus barrier, a physical barrier composed of intestinal epithelial cells (IECs) in the middle, and an immune defense layer in the inner layer [18]. Under normal circumstances, nutrients can be transported to the liver along this pathway. However, when the barrier function is impaired, harmful substances such as gut microorganisms and microbial products can also penetrate the barrier and transfer to the hepatic tissue, leading to liver inflammation [19,20].

Alcohol consumption has been indicated to change the microbial species within the gastrointestinal tract, causing injury to the gut mucosa barrier and permeability [20,21,22]. Normally, hepatic macrophages respond to small amounts of endotoxin from the intestine. However, due to increased intestinal permeability in ALD, microbes and microbial products are more readily translocated to the hepatic tissue, activating pattern recognition receptors (PRRs) on various cells in the hepatic tissue and triggering an inflammatory reaction [23]. In a normal hepatic organ, hepatic stellate cells (HSCs) remain quiescent, however, upon exposure to any liver damage, these cells become activated and subsequently transform into myofibroblasts. Specifically, myofibroblasts are recognized by increased proliferation, enhanced contractility, and heightened cytokine/chemokine production [24]. In addition, the activation of HSCs is essential in hepatic tissue because these cells secrete extracellular matrix and collagen for repairing liver damage. However, excessive production of collagen tends to promote liver fibrosis and sclerosis rather than simply repairing damage [25].

In conclusion, the integrity of the gut barrier is of paramount importance in preventing the translocation of intestinal microorganisms into the liver. Damage to the gut barrier provokes an increase in the transport of harmful substances like endotoxins. This not only exacerbates damage to the intestine itself but also promotes the inflammatory response and fibrosis process of the liver by activating liver macrophages and hepatic stellate cells. The excessive activation of liver macrophages and the transition of HSCs from a quiescent state act in concert to promote the progress of ALD and, in severe cases, may even transfer into stages of fibrosis and cirrhosis. A thorough understanding of gut–liver axis interactions and regulatory mechanisms is essential for developing effective ALD prevention and treatment strategies.

## 3. Alterations of the Gut Microbiota in ALD

The human gut contains more than 100 trillion microbial species [26]. The gut microbiota contains millions of bacterial genes when it comes to the human genome, which contains approximately 20,000 genes [27,28]. In the gut microbiota sequencing results, 99.9% of the sequences were identified as bacteria, while 0.01% were categorized as other classifications [29]. At the phylum level, *Firmicutes* and *Bacteroidetes* make up the greater part of the human gut microbiome [30]. Normally, the microbiome associated with humans exists in a symbiotic relationship, playing crucial roles such as digestion and metabolism, energy recovery, vitamin production, immune regulation, and epithelial cell differentiation [9,31]. However, the makeup and activity of gut microbes are shaped by many influences, such as diet, genetic code, age, and medication [32].

In general, alcohol consumption has been observed to result in a lower α-diversity of the human gut microbes [33], characterized by increased *Proteobacteria* and *Fusobacteria*, decreased *Firmicutes*, and an increase or decrease in the ratio of *Bacteroidetes* [14,29,33,34,35,36] (Table 1). Similar gut dysbiosis can even be seen in blood circulation [37]. On a genus basis, *Bifidobacteria*, as well as *Lactobacilli*, were evidently reduced in alcoholic patients compared with controls [11,22,33]. Compared with controls, the abundance of *Dorea* of the family *Lachnospiraceae* in ALD increased, while *Ruminococcus*, *Faecalibacterium*, and *anaerobic* bacteria decreased [22]. A clinical study noted a pronounced drop in *Lachnospiraceae* and *Ruminococcaceae* [34]. This study showed that, as opposed to the control group, the three most representative genera were *Bacteroides*, *Blautia*, and *Bifidobacterium*, and the reduced genera included *Prevotella*, *Paraprevotella*, and *Alistipes* in the gut microorganisms of patients with alcoholic cirrhosis [14].

In addition, studies have pointed out that changes in intestinal microbiota caused by alcohol-related diseases (such as chronic pancreatitis) need to be taken seriously [38,39]. A clinical study highlighted that chronic pancreatitis is connected with an intestinal microbes imbalance, known for augmented *Firmicutes* and reduced *Proteobacteria* levels [40]. Supplementation with bacteria like *Ruminococcus* that produce acetate, propionate, and butyrate can alleviate the severity of chronic pancreatitis, especially by reducing inflammatory damage [41]. However, larger-scale clinical trials are necessary in the future to test the safety and effect of gut microbiome treatment on extrahepatic symptoms in patients with ALD.

Interestingly, increased bacterial proliferation in the small intestine in individuals with chronic alcohol intake was first observed using traditional culture methods [42]. With the update of molecular ecological methods, numerous studies have found that variations in gut microbiome serve as indicators of disease severity [16,36,43]. For example, compared with healthy controls, the quantity of *Enterococcus faecalis* in the stool was increased in the group of AH, and cytolysin-positive *Enterococcus faecalis* was also found to be linked to the progression and death rate of ALD patients [43]. The more severe the alcoholic hepatitis, the lower the relative abundance of *Akkermansia* [36,44], whereas the increased relative abundance of *Veillonella*, unclassified *Clostridium*, unclassified *Prevotella*, *Bacteriaceae*, and *anaerobic* bacteria were lower [14,45]. Likewise, some studies have found higher proportions of *Bifidobacterium*, *Streptococcus*, *Enterobacter*, and *Enterococcus* in patients with severe AH [16,43]. More importantly, the combination of lowed *Akkermansia* and augmented *Bacteroidetes* was able to ascertain alcohol use disorder (AUD) patients with 93.4% accuracy [33]. Furthermore, a study found that *Atopobium* and *Clostridium leptospira* were inversely related to ALD progression [16]. In comparison to a person with non-alcoholic cirrhosis, those with alcoholic cirrhosis exhibit a greater frequency of endotoxemia [46], which is reasonably speculated to be related to the alterations in gut microbiota mentioned above. Therefore, these alterations can inspire us to target intestinal microbes for the treatment method of ALD.

**Table 1 biomedicines-13-00074-t001:** Alterations in the bacterial microbiota of ALD patients and animal models.

Participants	Comparison	Change in Gut Microbiota		Method	Ref
		Increased	Decreased		
Patients	ALD (n = 21) vs. HC (n = 16)	-	*Akkermansia muciniphila*	16S rRNA	Grander C, Adolph TE, Wieser V, et al. (2018) [47]
Patients	ALD (n = 14) vs. HC (n = 14)	*Alcaligenaceae*, *Rikenellaceae*, *Barnesillaceae*, *Paraprevotellaceae*, *Lachnospiraceae*	*Verrucomicrobiaceae*, *Bifidobacteriaceae*, *Akkermansia*, *Blautia*, *Bifidobacterium*, *Coprococcus* *Ruminococcus*	16S rRNA	Addolorato G et al. (2020) [33]
Patients	ALD (n = 19) vs. HC (n = 18)	*Clostridium*	*Bacteroides* *Bifidobacterium*	16S rRNA	Mutlu EA, Gillevet PM, Rangwala H, et al. (2012) [29]
Patients	AH (n = 70) vs. HC (n = 88)	*Enterococcus* *Escherichia coli*	-	16S rRNA	Duan Y, Llorente C, Lang S, et al. (2019) [43]
Patients	AUD (n = 10) vs. HC (n = 15)	*Lachnospiraceae* *Blautia*	*Ruminococcus*, *Faecalibacterium*, *Subdoligranulum*, *Oscillibacter* *Anaerofilum*	16S rRNA	Leclecrq S, Matamoros S etal. (2014) [22]
Patients	AH (n = 18) vs. SAH (n = 54)	*Unclassified Clostridales*, *unclassified Prevotellaceae*, *Anaerostipes*	*Akkermansia*	16S rRNA sequencing	Lang s, Fairfied B et al. (2020) [36]
Patients	ALD (n = 56) vs. HC (n = 20)	*Neisseriaceae*, *Chitinophagaceae*, *Bradyrhizobiaceae*, *Peptostreptococcaceae* *Turicellaand Microbacterium* *Anaerococcus*, *Lachnospiraceaeincertaesedis*, *ClostridiumXI*	-	16S rRNA sequencing	Puri et al. (2018) [37]
Patients	ALD (n = 27) vs. HC (n = 72)	*Klebsiella pneumoniae*, *Lactobacillus salivarius*, *Citrobacter koseri*, *Lactococcus lactis*	*Akkermansia*, *Coprococcus*, *unclassified Clostridiales*	16S rRNA sequencing	Dubinkina V et al. (2017) [14]
Patients	AH (n = 34) vs. HC (n = 24)	*Veillonellaceae*, *Proteobacteria*	*Lachnospiraceae*, *Ruminococcaceae*, *Porphyromonadaceae*, *Rikenellaceae*	16S rRNA sequencing	Smirnova E et al. (2020) [34]

AH, acute alcoholic hepatitis; SAH, severe alcoholic hepatitis; ALD, alcohol-related liver disease; HC, healthy controls.

## 4. The Possible Mechanisms That Gut Microbiota Promote the Development of ALD

### 4.1. Gut Microbiota and Intestinal Barrier

The intestinal barrier is disrupted in ALD, allowing pathogens and microbial products to be transferred into the circulation [16,22,48]. A large number of studies have observed that ALD models have endotoxemia, and those with endotoxemia are more severe than other causes of cirrhosis [33,35].

In the first place, gut microbiota dysbiosis, by reducing the production of antimicrobial proteins (AMPs) on the intestinal surface, promotes the destruction of the intestinal barrier. Such as C-type lectins and defensins, which play a bactericidal or antibacterial role [49,50]. Specifically, ethanol-related dysbiosis reduces the level of tryptophan metabolism into indole and the activation of the aryl hydrocarbon receptor (AHR) [51]. AHR inhibition reduces the expression of interleukin (IL)-22 (IL-22) and a C-type lectin named recombinant regenerating islet-derived protein 3 gamma (REG3G) in the intestine [51]. Mice fed a chronic alcohol diet have systemic bacterial dysbiosis accompanied by a decrease in α-defensins [52]. Knockout of function 7al α-defensins synergistically affects alcohol-induced alterations in bacterial species and gut barrier function, which worsens the phenomenon of translocation and liver injury [52].

Furthermore, α1,2-fucose is expressed at the top of IECs, and after glycosylation, it acts as a physical barrier for the host intestine [53]. Microorganisms regulate the glycosylation process. For instance, research on germ-free mice revealed that pups exhibited no intestinal fucosylation after being weaned from their mothers [54]. Gut commensal bacteria (such as *Bacillus* and *Ruminococcus*) promote fucosylation through IL-22 receptor, alpha1 (IL-22RA1) signaling by inducing the fucosyltransferase 2 (Fut2) [55]. We speculate that the destruction of the gut barrier in ALD may be altered by the reduction of commensal bacteria and impaired glycosylation, and more research will be required going forward. In addition, tight junctions between IECs are affected by short-chain fatty acids (SCFAs) because SCFAs perform as vital fuel substrates for IECs, thereby enhancing the assembly of tight junction proteins [48]. The SCFA-producing bacteria are reduced in ALD, which aggravates the destruction of intestinal tight junctions and bacterial translocation [48,56].

Finally, SCFA is crucial for sustaining intestinal immune homeostasis, which subsequently influences the health of the gut barrier. To be specific, they inhibit histone deacetylase (HDAC) and activate G-protein-coupled receptor 43/41 (GPR43/41, or named FFAR2/3) signaling pathways, effectively facilitating the activity of regulatory T cells (Tregs), thereby showing a protective effect against colitis in mouse models [57,58,59]. In addition, SCFAs reinforce the cytotoxic effectiveness of CD8^+^T cells, further strengthening the host’s ability to resist infection [60]. Apart from this, in a mouse model, researchers found that significantly reduced mucosal-associated invariant T (MAIT) cells abundance in mice barrier tissue was associated with alcohol exposure altering the gut microbiota [61]. MAIT cells contribute to antiviral and antibacterial defenses, tissue repair, and the maintenance of mucosal barrier integrity [62,63].

In summary, ALD caused by alcohol consumption is closely correlated to severe impairment of gut barrier function. This process involves gut dysbiosis and increased intestinal permeability. Ethanol-induced dysbiosis diminishes the production of helpful compounds like SCFAs, which are essential for protecting gut barrier integrity and immune function. Additionally, the reduction of distinct immune cells within the intestine, namely MAIT cells, further aggravates the damage to the intestinal barrier. Future research should investigate the role of intestinal microbial–host interactions in ALD etiology and explore treatments to restore intestinal barrier function and mitigate liver damage.

### 4.2. Gut Microbiota and Immunity

Current research reveals that inflammatory immune damage is vital in the evolution and escalation of ALD [64,65,66]. Inside the liver, maintaining normal liver homeostasis relies on a complex immune microenvironment composed of different immune cells and cytokines [67]. Especially in the context of ALD, intestinal barrier dysbiosis exacerbates the phenomenon of bacterial translocation. These bacterial components further activate the innate immune system through Toll-like receptors (TLRs) expressed by immune cells, triggering the release of inflammatory cytokines, the activation of immune cells, and the initiation of adaptive immune responses [68]. Previous research has shown that neutrophil infiltration is possibly a hallmark feature of ALD [69,70]. Evidence suggests that cytokines and chemokines like IL-17, IL-8, and monocyte chemotactic protein-1 (MCP-1) are instrumental in attracting neutrophils to the liver. Once these neutrophils infiltrate, they exacerbate alcohol-related liver fibrosis and hepatocyte apoptosis [71]. The specific mechanism involves human neutrophil peptide-1 (HNP-1) released by neutrophils, which further exacerbates alcohol-induced hepatic fibrosis and hepatocyte cell death by downregulating *anti-apoptotic factor B cell lymphoma 2(Bcl2)* expression and increasing *miRNA 34a-5p* expression [72]. After lipopolysaccharide (LPS) activates Kupffer cells (KC) through TLR4, growth factors like transforming growth factor-β (TGF-β) and platelet-derived growth factor (PDGF) are produced. These substances subsequently activate HSC, promote myofibroblast proliferation, generate extracellular matrix proteins, and ultimately accelerate the progression of hepatic fibrosis [73]. Simultaneously, the emission of pro-inflammatory cytokines by these macrophages, like MCP-1, is tightly connected to the early stages of alcoholic hepatic steatosis [74]. The manifestation of TGF-β promotes the differentiation of CD4+T cells into regulatory T cells, enhances their immunosuppressive function, and thus controls liver inflammatory responses [75]. A study pointed out that in AH, plasma endotoxin binds to CD14+ monocytes through TLR4, leading to overexpression of immunosuppressive receptors on T cells (such as programmed death receptor 1 (PD-1) and T cell immunoglobulin mucin domain protein 3 (TIM-3), weakening the body’s T cell function and accelerating the progression of disease [76]. Notably, microbiota dysbiosis in ALD leads to a significant dysbiosis of natural killer (NK) cells in the number and activity [13,77]. A marked decrease in the amount of NK cells in the hepatic tissue was also noted in an animal model that was given a chronic ethanol diet [78]. Impairment of this cytotoxic function shows a positive relationship with the magnitude of liver damage [79]. Researchers have pointed out that NK cell activation delays the progression of liver fibrosis by inhibiting HSC [80]. However, while probiotic supplementation in the mice was shown to alleviate hepatic inflammation and repair gut barrier damage in alcohol-fed mice, elevation in NK cell activity was also observed [13]. Therefore, we speculate that the reduction of NK cells aggravates ALD progression.

In addition, the recent literature also shows that the intestinal microbiota can actively manipulate pro-inflammatory and anti-inflammatory cytokines [81,82]. LPS activates KC cells and monocytes to release abundant inflammatory mediators. For example, factors with pro-inflammatory properties include tumor necrosis factor α (TNF-α), IL-1, IL-8, IL-18, and anti-inflammatory factors such as IL-10, IL-6, TGF-β. Polysaccharide A from Bacteroides promotes regulatory T cell differentiation and IL-10 secretion, leading to an anti-inflammatory response [83]. In patients with alcoholic hepatitis, after supplementation with *Lactobacillus rhamnosus* GG, the patient’s serum TNF-α and IL-1β levels decreased, and IL-10 amounts increased [11]. Restoring the gut microbiota of ALD patients helps to increase anti-inflammatory factors and reduce inflammatory damage.

In summary, an important contributor to the pathogenesis of ALD is inflammatory immune breakdown. Bacterial translocation triggered by decreased intestinal barrier function activates the liver’s innate immune system, triggering a range of complex immune responses. The infiltration of neutrophils and hepatic macrophages, along with the cytokines they secrete, significantly exacerbates liver fibrosis and hepatocyte apoptosis. Simultaneously, T cell and NK cell activity dysfunction also promote the progression of ALD. Intestinal microorganisms may improve ALD through modulating the activity of immune cells, providing new perspectives for probiotics as a treatment strategy for ALD.

### 4.3. Gut Microbiota Regulate Receptors and Signaling Pathways

Further exploring the mechanism of how gut microbes promote the advancement and progression of ALD, a core link is that gut microbiota accelerates the disease process through the receptor signaling pathway network (Figure 3). Immunogenic substances translocated from the intestine to the liver can activate PRRs, thereby inducing a strong inflammatory response [84,85]. Among PRRs, TLRs, as important receptors for recognizing pathogen-associated molecular patterns (PAMPs) and damage-associated molecular patterns (DAMPs), are particularly crucial in ALD [86]. To date, TLR10 and TLR12 have been recognized in humans and mice, in turn. Among them, TLR1-TLR9 are highly preserved across both organisms, while TLR10 in mice is non-functional, and TLR11, TLR12, and TLR13 are missing in the human genome [68,84]. These TLRs are present on different cell types in the hepatic tissue, for instance, KC, endothelial cells, dendritic cells, HSCs, and hepatocytes. In particular, TLR4, TLR5, TLR2, TLR3, and TLR9 recognize LPS, flagellin, lipoteichoic acid, RNA, and DNA, respectively, triggering inflammatory signaling mechanisms [74,87,88]. Among them, TLR4 is extensively acknowledged and investigated in the context of alcohol-induced liver inflammation [89]. Upon LPS binding to TLR4, the MyD88-dependent pathway activates nuclear factor κB (NF-κB) and mitogen-activated protein kinases (MAPKs), causing the generation of proinflammatory cytokines and chemokines, including TNF-α and IL-6 [90]. On the other hand, LPS activates interferon regulatory factor 3 (IRF3) through the TIR domain-containing adaptor factor (TRIF) dependent pathway, prompting the generation of type I interferons (like IFN-β) [91,92]. Indeed, mice lacking TRIF or mice lacking IRF3 are protected from ALD [93,94]. Evidence suggests that TLR4 signaling in ALD primarily occurs via a MyD88-independent pathway. Moreover, in ALD, NOD-like receptors (NLRs), as another type of PRR, are also important. They assemble inflammasomes with the adaptor protein ASC and the effector protein procaspase-1 [95]. The NLRP3 inflammasome is particularly pivotal in ALD. Upon activation by PAMPs and DAMPs, it triggers caspase-1, which cleaves cytokine precursors like interleukin-1β (IL-1β) and interleukin-18 (IL-18), resulting in the production of mature cytokines [96]. Interestingly, there is a crosstalk between the TLR pathway and oxidative stress in ALD. The LPS/TLR4 receptor complex activates NADPH oxidase in KC, leading to the production of reactive oxygen species (ROS). ROS not only directly damages liver cells but also activates KC and induces them to secrete more TNF-α [97]. TNF-α, through its receptor tumor necrosis factor receptor 1 (TNF-R1), initiates a series of downstream signaling pathways, promoting liver inflammation and hepatocyte apoptosis [98]. The mechanism involves activating transcription factors like NF-kB and c-Jun-N-terminal kinase, as well as initiating the pro-apoptotic Fas-associated death domain [99,100]. TNF-α also affects lipid metabolism, increasing the expression of the hepatic transcription factor SREBP-1c to promote lipid synthesis in mouse and human hepatocytes [101,102]. In ALD, IL-6 has dual roles, exhibiting both pro-inflammatory and anti-inflammatory effects [103,104]. IL-6 enhances pro-inflammatory cytokine expression in KC, contributing to its pro-inflammatory effects [104]. IL-6 and STAT3 modulate liver inflammation in a cell type-specific manner: STAT3 in hepatocytes enhances inflammation, while STAT3 in macrophages/KC suppresses it [103]. Some researchers pointed out that IL-6 is highly correlated with liver performance in individuals with AUD and may help identify high-risk patients with poor mid-term prognosis, suggesting its potential value in the treatment and prognosis assessment of AUD [105]. Bicyclol exerts its hepatoprotective effect primarily by decreasing IL-6 production and inhibiting the IL-6/STAT3 signaling pathway activation [106].

In addition to the above methods, gram-negative bacteria also deliver their contents (including LPS) into host cells by secreting LPS-loaded outer membrane vesicles (OMVs) [107]. This suggests that bacteria can allow LPS to enter the cytosol without going through the invasion process, thereby activating cytoplasmic caspase-11/4 and leading to gasdermin D-mediated pyroptosis. Cytoplasmic caspase-11 (also known as caspase-4) characterizes the non-classical inflammasome triggered by infections with multiple Gram-negative bacteria, leading to programmed cell death via pyroptosis [108,109]. This pattern was also demonstrated to be associated with the progression of ALD [110].

In summary, an extensive amount of immunogenic substances enter the hepatic tissue through the damaged intestinal barrier, activate PRRs on KC, hepatocytes, and immune cells, and induce a strong inflammatory response. Long-term inflammatory response leads to functional and structural damage to the liver, thereby triggering the progression of ALD. The persistent production of cytokines and chemokines, including TNF-α and IL, is vital in liver fibrosis development. Bicyclol enhances liver function by targeting the IL-6/STAT3 pathway, highlighting the significance of signaling pathways. This indicates that managing gut microbiota, limiting intestinal content translocation, and reducing inflammatory cell activation may serve as potential therapeutic targets for ALD.

### 4.4. Gut Microbiota Regulate Metabolites

The gut microbes of AH patients showed significant structural and functional changes compared to non-alcoholic controls, particularly significant differences in functional metagenomes, serum, and fecal metabolites [111]. Changes in metabolites are very important in the pathogenesis of ALD, and their alterations are considered to be harmful to liver disease [112]. Metabolite disturbances were also found in animal models [113]. Key microbial metabolites include bile acids, SCFAs, branched-chain amino acids (BCAAs), and amino acid metabolism products (like trimethylamine-N-oxide (TMAO) and trimethylamine (TMA)) [114].

## 5. Bile Acids

Intestinal microbial bile salt hydrolase (BSH) converts bile acids into free bile acids, which are then metabolized by bacterial 7α-dioxygenase into secondary bile acids [115]. In ALD, there is a gut dysbiosis responsible for BSH production (such as *Lactobacillus*, *Bifidobacterium*, and *Bacteroidetes*) [33,116]. Bile acids are synthesized by the individual but metabolized by intestinal microbes. ALD is accompanied by disruption of bile acid metabolism, manifested by increased systemic total bile acid and suppressed hepatic bile acid synthesis [117,118]. Farnesoid X receptor (FXR) signaling is crucial for regulating bile acid, glucose, and lipid metabolism, impacting energy homeostasis and metabolic functions. In the human intestine, the binding of primary bile acids to FXR activates a signaling pathway that leads to the production of fibroblast growth factor 19 (FGF19) [119]. Subsequently, FGF19 can enter the liver through the gut–liver axis. FGF19 suppresses cholesterol 7 alpha-hydroxylase (CYP7A1) in the liver, decreasing bile acid synthesis and enhancing their enterohepatic circulation [120]. In persons with ALD, intestinal FXR levels are decreased, while total and conjugated bile acids are significantly elevated in those with AH, indicating dysregulation of bile acid homeostasis and related signaling pathways [121,122]. Similarly, in mice fed the Lieber DeCarli ethanol diet, FXR activity and fibroblast growth factor 15 (FGF 15) expression in IECs were reduced, leading to an increase in liver CYP7A1 levels and increasing blood bile acid concentrations [123]. This is connected with raised levels of unconjugated bile acids in the gut due to alcohol-induced intestinal dysbiosis. Depletion of FXR renders mice susceptible to hepatic steatosis and ethanol-induced liver damage [121]. Interestingly, FXR signaling also affects multiple antimicrobial agents, including angiopoietin 1 and RNAse family members 4, and the decrease in these bactericidal proteins is correlated with the occurrence of small intestinal bacterial overgrowth in mice [124,125]. Takeda G protein-coupled receptor 5 (TGR5, also known as GPBAR1) is another bile acid-responsive receptor present on various cells, including KC, immune cells, and adipose connective tissue [21,126]. Bile acids mitigate liver injury by suppressing LPS-induced cytokine expression in KC through a TGR5-cAMP-dependent mechanism [127]. The study observed a reduced abundance of bacterial genes related to bile acid conversion in alcohol-fed TGR5-deficient mice compared to WT mice [127]. A recent study has shown that TGR5-deficient mice display more severe steatosis and inflammation than wild-type mice [128]. Thus, targeting gut microbiota and TGR5 could be beneficial for addressing alcohol-induced liver injury in humans. Overall, the concentrations of bile acids in the feces and serum of cirrhotic patients who drink alcohol are increased, especially the secondary bile acids are significantly increased [45,129,130]. Elevated secondary bile acids (DCA) can activate HSCs, heightening the risk of HCC development [131]. However, in advanced cirrhosis, serum levels of conjugated bile acids are increased while total bile acid levels are decreased [45]. This alteration in the bile acid pool resembles earlier findings and correlates with disease severity [132].

Therefore, enhanced bile acid testing can help identify disease progression. It can be seen that manipulating the gut microbiome and bile acid metabolism is expected to become an effective strategy for dealing with ALD.

## 6. Short-Chain Fatty Acid

Short-chain fatty acids, generated through bacterial fermentation of intestinal carbohydrates, are crucial for energy supply and homeostasis regulation in the gut and other organs [133,134]. Acetic, propionic, and butyric acids are the predominant SCFAs in the gut [133]. Ethanol consumption generally reduced SCFA levels in the gut, except for acetic acid, which increased alongside ethanol metabolites [113]. Increased acetic acid levels post-alcohol consumption may arise from ethanol’s oxidation to acetaldehyde, followed by its conversion to acetic acid [135]. Due to limited bacterial aldehyde dehydrogenase activity [136], the gut microbes are not a primary contributor to the increase in luminal acetate levels. A recent study reported that stool specimens from AH patients contained lower levels of SCFAs compared with the control groups. Consistently with earlier studies, researchers have found that long-term alcohol consumption diminishes gut concentrations of SCFAs [137].

Study shows that SCFAs can counteract alcohol-induced gut barrier dysfunction by activating AMP-activated protein kinase (AMPK) in colorectal adenocarcinoma (Caco-2) cells [138]. Butyrate has been shown to enhance mucin secretion and strengthen the intestinal barrier [139]. A study in mice demonstrated that butyrate supplementation mitigated the harmful effects of alcohol consumption on gut-tight junctions [140]. This shows that SCFAs enhance the gut barrier and decrease the risk of endotoxin transfer to the liver.

SCFAs, like propionic acid and butyric acid, bind to GPR43 and GPR41 on immune cells and IECs, inhibiting the phosphorylation of lipid MAPKs as well as the activity of NF-κB, thereby downregulating inflammatory factors (like inducible nitric oxide synthase (iNOS), TNF-α, MCP-1 and IL-6) reduce immune cell recruitment and exert anti-inflammatory effects [141,142]. However, acetate is a key substrate for histone acetylation and promotes histone acetylation by increasing the generation of acetyl-CoA, thereby enhancing the transcriptional activity of these genes [143]. This mechanism suggests that acetate increases the proinflammatory response of macrophages by regulating epigenetic modifications. Overall, SCFAs exhibit both anti-inflammatory and pro-inflammatory functions, and further research is essential to fully understand the mechanisms through which they exert these effects, particularly in the context of ALD. Although the majority of SCFAs are generated and exert their effects primarily within the intestine, certain SCFAs, including acetate and propionate, have the ability to travel to and affect the liver. In the liver, SCFAs serve as raw materials for the synthesis of lipids and glucose, regulating glucose homeostasis and affecting lipid metabolism [142]. Butyrate activates the AMPK signaling pathway to regulate liver lipid metabolism and decrease liver fat accumulation [144]. Acetate plays a role in the synthesis of cholesterol and fats, while propionate has an inhibitory effect on this process [139,145]. However, acetate increased, and other SCFAs decreased in ALD, and this change in the ratio of SCFAs may be an important factor promoting alcoholic hepatic steatosis. Interestingly, SCFA content was associated with gut microbiota. In patients with AUD, a negative correlation was observed between alcohol dependence and both the relative abundance of butyrate-producing species in the order *Clostridiales* and the opportunistic pathogen *Enterobacteriaceae* [14]. The abundance of *Proteobacteria* was negatively correlated with SCFA content [146]. The abundance of *Faecalibacterium* was positively correlated with butyrate concentrations and negatively correlated with isobutyrate and isovaleric acid [146].

In conclusion, the reduction of SCFAs in ALD has been found to impair the functionality of the gut barrier and promote hepatic steatosis and inflammation. Gut microbiota dysbiosis in drinkers may lead to changes in intestinal fermentation processes, affecting the metabolism of SCFAs and thereby promoting the progression of ALD.

## 7. Amino Acids

Branched-chain amino acids are essential nutrients acquired through diet because they require the participation of gut microbes [147]. Mice fed with a chronic ethanol diet over an 8-week period significantly resulted in reductions among almost all amino acids, including all three BCAAs (leucine, isoleucine, valine) [148]. BCAAs have been demonstrated to alleviate liver fibrosis and reduce the risk of tumor formation in mice subjected to a high-fat diet. This protective effect is mediated through the suppression of PDGFR-β/ERK signaling pathways, which play a critical role in fibrogenesis and oncogenesis [149]. BCAA supplementation may improve liver disease severity [150]. This suggests that dysregulation of BCAAs homeostasis may possibly be one of the reasons for liver disease.

This suggests that dysregulation of BCAAs homeostasis may possibly be one of the mechanisms of liver disease. Dietary tryptophan is metabolized by the gut microbiota into indole and its derivatives, including indole-3-acetic acid (IAA) and indole-3-propionic acid (IPA) [151]. Indole is an aryl hydrocarbon receptor ligand that can stimulate RORγt type 3 innate lymphoid cells (ILC3) to produce IL-22, thereby inducing the production of antimicrobial proteins [51]. IPA indirectly inhibits hepatic NF-κB signaling, thereby significantly reducing hepatic inflammation and liver injury [146]. Indole levels are reduced in stool samples of patients with AH [51,146,152]. This further reduces IL22 expression in the intestine, leading to decreased expression of the antimicrobial REG3G, which in turn results in increased bacterial translocation. Bacteria engineered to produce IL22 induce REG3G expression and reduce ethanol-induced steatohepatitis [51]. Furthermore, gut bacterial fermentation of proteins and amino acids produces excess ammonia (a toxic compound) [153]. In the context of gut dysfunction, these toxic byproducts contribute to the onset of liver-related complications (like hepatic encephalopathy) [154]. In summary, the imbalance between amino acid metabolism and AHR promotes the progression of ALD. Restoring this imbalance may help improve liver damage and provide a new way to manage ALD.

## 8. Other Metabolites

In addition to the changes in the above-mentioned metabolites, some metabolites have been pointed out as markers of ALD, and early identification may help us diagnose and treat them earlier. A study indicates that serum 3-ureidopropionate, cis-3,3-methyleneheptanoylglycine, retinol, and valine levels may be used as biomarkers in the clinical assessment of alcohol-related cirrhosis [155]. Palmitoylcarnitine may be a potential promising biomarker for diagnosing ALD [146]. Gut microbes can metabolize choline into TMA. A study has pointed out that TMA is increased in AH participants compared with healthy controls [156]. TMA can be further oxidized by liver monooxygenase to TMAO. The study highlighted that as ALD advances, the rise in TMAO and its precursors, such as carnitine, correlates with the degree of the disease. The study highlighted TMAO as a dependable biomarker for monitoring ALD progression [157]. Researchers treated mice with an inhibitor of bacterial choline TMA lyase (CutC/D), which protected them from ethanol-induced liver damage [156]. This implies a potential link between TMA, TMAO, and the development of ALD. Furthermore, metabolites like cytolysin from *Enterococcus faecalis* and cyanidins from *Candida* influence host metabolism and outcomes [43,158]. The researchers found that cytolysin is a more effective method for identifying mortality from ALD within 90 days compared to other methods [43].

## 9. Therapy Methods for ALD Target Gut Microbiota

Alterations within the species of gut microbes, microorganism-based metabolites, and gut barrier dysfunction are contributors leading to the advancement of ALD. Lately, researchers have focused on the study of the microbiome in liver disease treatment. Non-targeted therapies, such as probiotics, prebiotics, and FMT, are utilized. Targeted therapies encompass bacteriophages and genetically engineered bacteria. These methods can alter the gut microbe composition in ALD patients, restoring microbial balance, preventing bacterial translocation, and improving liver damage. This article mainly discusses the research results related to therapies such as probiotics, FMT, and phages (Table 2 and Appendix A).

At the same time, some researchers have pointed out that the intake of red wine composed of different polyphenols (such as resveratrol) and the Mediterranean diet can prolong life [170]. In particular, resveratrol contained in wine seems to have a positive effect on longevity [170]. The specific mechanism is that it has antioxidant, anti-inflammatory, and anti-cancer properties. A cohort study also pointed out that the Mediterranean drinking pattern does not increase the incidence of atrial fibrillation [171]. This suggests that low to moderate drinking may not affect our health. In addition, polyphenols in distilled spirits can also improve ALD [172]. Therefore, the impact of alcohol consumption on health and ALD needs further study.

## 10. Probiotics

Probiotics, a class of beneficial microorganisms that are not harmful, offer health advantages to humans. Probiotics function by normalizing the intestinal microbiota, competitively excluding pathogens, producing bacteriocins, modulating the immunological system as well as participating in intestinal energy metabolism [173,174,175]. They aid in reversing gut dysbiosis and reducing inflammation from microbial translocation, offering a potential treatment for ALD [176,177,178]. A randomized controlled trial showed that cirrhosis patients receiving probiotics had a microbiome enriched with strains like *Faecalibacterium prausnitzii*, *Syntrophococcus sucromutans*, *Bacteroides vulgatus*, *Alistipes shahii*, and a *Prevotella species*, compared with the placebo group [159]. Research indicates that probiotic treatment in alcoholic patients can restore suppressed *Bifidobacteria*, *Lactobacilli*, and *Enterococci* to levels comparable to healthy individuals, correlating with improved liver enzyme function [11]. Patients with alcoholic liver injury showed augmented *Lactobacillus* and *Bifidobacterium* and lowered triglyceride proportions after receiving *Lactobacillus casei strain Shirota* when contrasted with the control group [160]. Research indicates that the probiotic *Lactobacillus casei Shirota* may restore neutrophil phagocytosis in ALD by altering IL10 secretion and TLR4 expression and decreasing inflammatory cytokines in vivo [161,162]. There are also some studies that indicate that probiotics such as *Bifidobacterium breve* ATCC15700, *Escherichia coli Nissle*, *Lactobacillus subtilis*, and *Streptococcus faecalis* can restore intestinal barrier function and normalize the structure and composition of the gut microbiota damaged by alcohol, thereby significantly reducing endotoxemia, maintaining immune homeostasis, and alleviating alcohol-induced liver damage [163,164,179]. In addition, similar findings regarding the effects of probiotics have been found in animal models. In the mouse model of ALD, supplementation with VSL#3 modulated the ecological balance of the gut microbes, blocked the entry of endotoxins and other microbial-derived compounds from the intestinal lumen into the portal circulation, and downregulated the expression of TNFα [180]. *Lactobacillus rhamnosus Gorbach-Goldin* probiotic gavage significantly improved intestinal and liver oxidative stress in mice, reduced liver macrophage infiltration, reduced inflammatory markers, and restored intestinal barrier function [181,182,183,184]. Many studies have also shown that *Lactobacillus rhamnosus* R0011, *Lactobacillus rhamnosus* NKU FL1-8, and *Lactobacillus acidophilus* R0052 can reduce IL-1β, TNF-α, and LPS, thereby reducing liver inflammation [185,186,187]. In alcohol-fed mice, supplementation with *Phocaeicola dorei* and *Lactobacillus helveticus* increased NK cell activity [13]. Studies have shown that administration of *Akkermansia muciniphila* prevented neutrophil infiltration and improved liver damage and steatosis in mice with ALD [47]. *Lactobacillus reuteri* was able to convert the dietary component L-histidine into the immunomodulatory signal histamine, thereby inhibiting the production of pro-inflammatory TNF and improving hepatic lipid accumulation in mice with ALD model through the FXR signaling axis [188,189]. *Pediococcus pentosaceus* CGMCC 7049 mitigated liver damage in mice fed alcohol by restoring gut dysbiosis, enhancing the gut barrier, and lowering circulating endotoxin levels as well as inflammatory factors [190].

In conclusion, probiotic supplementation has achieved good results in both ALD patients and mouse models, mainly by normalizing species of the microbiome and restoring the gut barrier, subsequently lowering liver inflammation and greater function. More large-scale clinical trials are needed in the future to verify this effect.

## 11. FMT

Fecal microbiota transplantation, a procedure in which stool from a healthy donor is introduced into the patient’s digestive tract, entails the transfer of an entire microbial community. In murine models, FMT from alcohol-resistant to alcohol-sensitive mice has been shown to prevent *Bacteroidetes* reduction and steatosis development [15]. Similarly, FMT from healthy humans is an efficient treatment for ALD, enhancing gut microbiota diversity and beneficial bacteria abundance [165,191]. In a study on severe AH patients undergoing FMT, a week post-treatment, their intestines showed a dominance of less pathogenic bacteria such as *Bacteroidetes*, *Parabacteroidetes*, and *Porphyromonas* [191]. Another study reported that patients with ALD showed huge improvements in liver damage and survival within one week after FMT, accompanied by changes in intestinal metabolic pathways [166], such as increased synthesis of SCFAs [165,166,167,168].

Furthermore, research has demonstrated that the oral administration of FMT capsules, which are rich in *Lachnospiraceae* and *Ruminococcaceae*, can enhance the gut microbes of patients with ALD and reduce serum lipopolysaccharide-binding protein (LBP). LBP facilitates the binding of LPS to its receptor by interacting with the lipid A component of LPS.

Additionally, researchers conducted a comparative analysis of nutritional therapy, corticosteroids, pentoxifylline, and FMT under different treatment modalities. FMT demonstrated superior efficacy in improving the survival rate of patients with SAH compared to the survival rate achievable with existing treatment modalities [191]. A follow-up of patients with SAH demonstrated significantly elevated 28-day and 90-day survival rates, in addition to improved clinical severity scores, in the FMT cohort relative to the standard care therapy group [169].

In conclusion, studies on FMT use in AH and cirrhosis patients show promising results. Clinical indicators and survival rates of ALD patients were improved after FMT, especially in individuals with SAH.

## 12. Bacteriophage

Bacteriophages are viruses that target and destroy bacteria. They are progressively regarded as potential treatments for various diseases, including ALD. Bacteriophages can be engineered or selectively targeted to specific bacteria in the gut to mitigate the effects of alcohol on the liver by lysing pathogens and reducing endotoxin levels and inflammation [192]. A recent study showed that the presence of cytolysin-bearing *Enterococcus faecium* is positively associated with the severity of liver disease and mortality in patients. In humanized mouse models, phages targeting cytolysin-positive *Enterococcus faecium* significantly reduced cytolysin and effectively alleviated ALD symptoms [43]. In conclusion, targeted phages provide a new perspective for the treatment of ALD, demonstrating the potential of phages in improving liver function by precisely targeting specific intestinal pathogens. The research on bacteriophages is still insufficient, and more research is needed to provide evidence.

## 13. Future and Prospects

A substantial body of research has demonstrated that the species of the gut microbiome in patients with ALD has undergone alterations. Additionally, notable discrepancies have been observed in the gut microbes between individuals with severe ALD and those with mild ALD. The rise in pathogens and the decline in beneficial bacteria result in gut barrier dysbiosis, increasing the contact between gut submucosal immune cells and immunogenic compounds. This, in turn, leads to an increase in inflammation and a further deterioration in intestinal permeability. Damage to the intestine has been demonstrated to significantly enhance the inflammatory and immune responses of the liver. On the one hand, this response facilitates the clearance of pathogens and their metabolites. Conversely, sustained chronic inflammatory processes also exacerbate the pathological alterations of the liver, ultimately leading to SAH, liver fibrosis, and even cancer. This significantly impacts the prognosis of patients. This review also introduces the therapeutic use of probiotics, FMT, and targeted phages to restore the gut microbiome, modulate immune and inflammatory responses, and improve metabolic disorders, thereby alleviating alcohol-induced liver injury.

Future research still needs to make efforts in the following aspects: Vertically, it is essential to continue to dig deeper into the mechanisms of pathogenic strains, especially identify strains exerting a real impact on ALD progression through multi-center and large sample-based clinical trials, and find the key pathogenic genes and microbes-derived metabolites through multi-omics technologies, as well as clarify their impacts on gut barrier function and host immunity, etc. It is also urgent to demonstrate horizontally the interactions between varying microbes, including the mechanisms of probiotics inhibiting harmful bacteria, phage targeting the removal of bad microbes, and so on. As for phage removal of gut fungi like *Candida*, further research is still needed. In addition, even harmful bacteria are not always bad, which may be beneficial to health after modification, such as *Candida* albicans-related training immunity. Pre-administration of relevant strains to alcoholics to produce a vaccine-like response in host immunity may have a preventive effect on the subsequent progression of ALD. Finally, the research technology regarding microbiota still needs to be improved, such as the use of sterile animals and gut–liver organ microarrays, which will help to deepen our understanding of the above mechanisms and provide high-level research evidence.

## Figures and Tables

**Figure 1 biomedicines-13-00074-f001:**
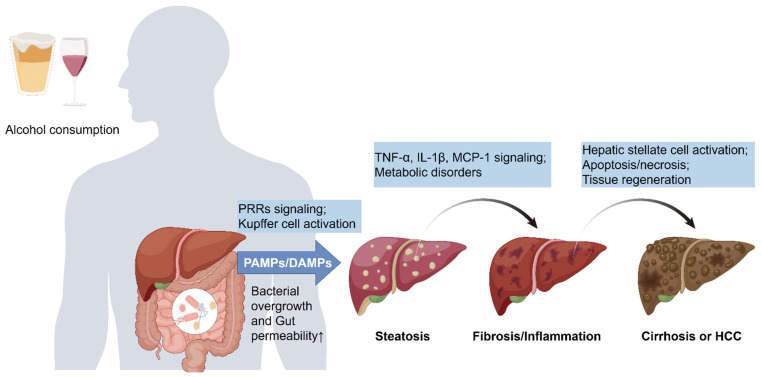
Mechanism of alcohol-related liver disease damage caused by gut microbiota. Drinking alcohol can lead to intestinal microbiota imbalance and increased intestinal permeability. A large number of immunogenic substances enter the circulatory system and reach the liver, activating PRR receptors on various cells in the liver and triggering the release of a large number of cytokines and chemokines. These substances accumulate in the liver, leading to changes in liver metabolism, inflammation, fibrosis and cirrhosis, and even the occurrence of hepatocellular carcinoma. PAMPs, pathogen-associated molecular patterns; DAMPs, damage-associated molecular patterns; PRRs, pattern recognition receptors; TNF-α, tumor necrosis factor α; IL-1β, interleukin-1β; MCP-1, monocyte chemotactic protein-1; HCC, hepatocellular carcinoma.

**Figure 2 biomedicines-13-00074-f002:**
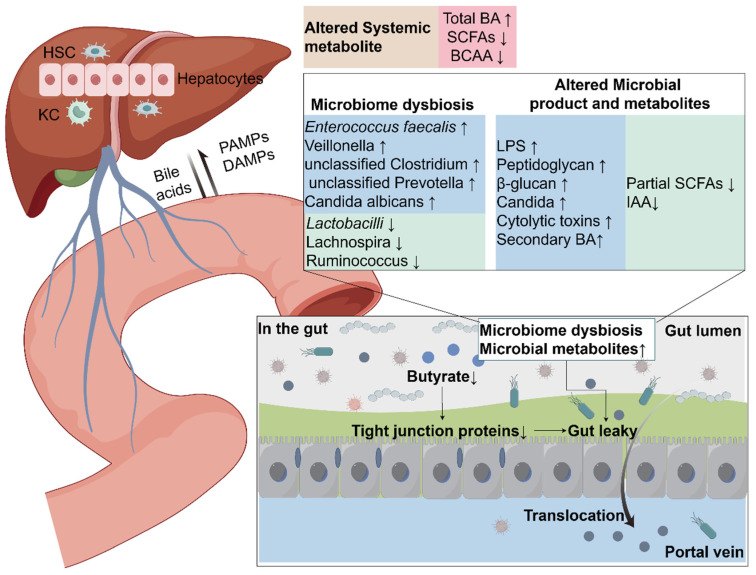
Gut–liver axis in ALD. Alcohol consumption leads to changes in intestinal microbiota, which in turn affect changes in metabolites. Multiple factors lead to increased intestinal permeability, especially the reduction of butyrate, which destroys intestinal tight junctions and promotes the translocation of bacteria and their products into the portal circulation. Ultimately, continued infiltration of immunogenic substances and activation of various cells in the liver will lead to liver fibrosis and cirrhosis. The gut–liver axis promotes inflammation, metabolic changes (such as bile acid metabolism), and cell death in ALD. HSC, hepatic stellate cells; KC, Kupffer cells; PAMPs, pathogen-associated molecular patterns; DAMPs, damage-associated molecular patterns; Total BA, bile acids; SCFAs, short-chain fatty acids; BCAA, branched-chain amino acids; LPS, lipopolysaccharide; IAA, indole-3-acetic acid.

**Figure 3 biomedicines-13-00074-f003:**
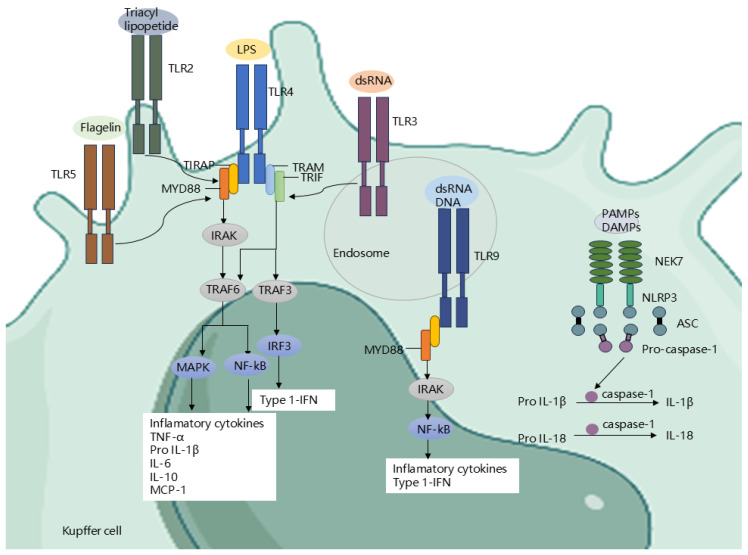
Receptor pathways in the liver. Activation of NLR by PAMPs and DAMPs. Activation of NLRP3 inflammasomes, thereby promoting caspase-1 cleavage and IL-1β and IL-18 secretion. LPS activates TLR4 expressed on Kupffer cells, HSCs, and hepatocytes to initiate IRF3, MAPK, and NF-kB signaling pathways to enhance the secretion of inflammatory chemokines and cytokines. In addition, bacterial DNA and RNA activate TLR9 and TLR3 on endosomes, thereby activating the NF-kB/MAPK signaling axis and subsequently secreting inflammatory cytokines. In addition, other immunogenic substances, such as flagellin and triacyl lipopeptide, activate TLR5 and TLR2 on KC to promote the secretion of inflammatory factors. PAMPs, pathogen-associated molecular patterns; DAMPs, damage-associated molecular patterns; NLRs, NOD-like receptors; NLRP3, NOD-like receptor protein 3; NEK7, NIMA-related kinase 7; ASC, apoptosis-associated speck-like protein containing a CARD; LPS, lipopolysaccharide; TLR4/9/3/5/2, Toll-like receptor 4/9/3/5/2; MyD88, myeloid differentiation factor 88; TRIAP, TIR domain containing adaptor protein; IRAK, IL-1R-associated kinase; NF-kB, nuclear factor kappa B; MAPK, mitogen-activated protein kinase; TRAM, Trif-related adaptor molecule; TRIF, adaptor containing TIR domain that induces IFN-b; RIP3, receptor-interacting protein kinase 3; TNF-a, tumor necrosis factor-a; IL-1β, interleukin-1β; IL-18, interleukin-18; IL-6, interleukin-6; IL-10, interleukin-10; MCP-1, monocyte chemotactic protein-1; Type 1-IFN, Type-I interferons.

**Table 2 biomedicines-13-00074-t002:** Overview of the potential therapeutic approaches in ALD.

Intervention/Therapy	Subjects	Dose and Method	Result and Conclusion	Study Design
Probiotics	Patients with compensated cirrhosis	1.5 × 10^10^ CFU for 6 months; oral administration.	An increase in probiotic bacteria and other beneficial taxa in stool of compensated cirrhotic patients.	Clinical study
*Bifidobacterium bifidum* W23, *Bifidobacterium lactis* W51, *Bifidobacterium lactis* W52, *Lactobacillus acidophilus* W37, *Lactobacillus brevis* W63, *Lactobacillus casei* W56, *Lactobacillus salivarius* W24, *Lactococcus lactis W19* and *Lactococcus lactis* W58 (Horvath A et al., 2020) [159]				
*Bifidobacterium bifidum*, *Lactobacillus plantarum* 8PA3 (Kirpich I A et al., 2008) [11]	Patients with alcohol-induced liver injury	0.9 × 10^8^ CFU for five consecutive days; oral administration.	A significant end of treatment reduction in ALT, AST, GGT, LDH and total bilirubin. A significantly increased numbers of both bifidobacteria and lactobacilli.	Clinical study
*Lactobacillus casei strain Shirota* (Li X et al., 2021) [160]	Patients with alcoholic liver injury	100/200 mL of *Lactobacillus casei* strain Shirota (low-dose group/high-dose group) for 60 days; oral administration.	A significantly decrease in the serum levels of TG and LDLC. A significant increase in the amount of *Lactobacillus* and *Bifidobacterium.*	Clinical study
*Lactobacillus casei Shirota* (Stadlbauer V et al., 2008) [161]	Patients with alcoholic cirrhosis	6.5 × 10^9^ CFU for 4 weeks; oral administration	A capacity about probiotics restore neutrophil phagocytic in cirrhosis	Clinical study
*Lactobacillus casei Shirota* (J M et al., 2020) [162]	Patients with liver cirrhosis	6.5 × 10^9^ CFU for 6 months; oral administration.	An improvement about cytokine profile towards an anti-inflammatory phenotype.	Clinical study
*Escherichia coli Nissle* (Mutaflor) (Lata J et al., 2007) [163]	Patients with liver cirrhosis.	2.5–25 × 10^9^ CFU for 42 days; oral administration.	Significantly restored intestinal microflora including *Lactobacilli* and *Bifidobacteria*. Reduced endotoxemia and potentially improved liver function.	Clinical study
*Lactobacillus subtilis*/*Streptococcus faecium* (Han S H et al., 2015) [164]	Patients with alcoholic hepatitis (AH)	1500 mg/day for 7 days (1500 mg/day); oral administration	The restoration of the bowel flora and improvement in LPS in patients with AH.	Clinical study
FMT from a donor enriched in *Lachnospiraceae* and *Ruminococcaceae* (Bajaj J S et al., 2021) [165]	Patients with AUD-related cirrhosis with problem drinking	90 mL (27 g of stool) of the FMT material from the donor containing approximately 2.7 × 10^12^ CFU; via enema	Safe; favorable microbial changes.	Clinical study
FMT (Philips C A et al., 2017) [166]	Patients of ALD	30 g donor stool samples homogenized with 100 mL of sterile normal saline in a blender, filtered using sterile gauze, and small aliquots infused through a nasoduodenal tube daily for 7 days.	Improved significantly liver disease severity within the first week after FMT; proteobacteria reduced and Actinobacteria increased. Improved survival rate.	Clinical study
FMT from a donor enriched in *Lachnospiraceae* and *Ruminococcaceae* (Bajaj J S et al., 2018) [167]	Patients with cirrhosis (exposed to multiple courses of antibiotics)	A 90 mL of FMT from one donor or to standard of care; via enema.	Safe and well-tolerated. Restored antibiotic-associated disruption in microbial diversity and function.	Clinical study
FMT (Bajaj J S et al., 2019) [168]	Patients diagnosed with cirrhosis and recurrent hepatic encephalopathy	15 oral capsules containing healthy donor FMT; oral administration.	Safe and well-tolerated; significantly increased duodenal microbial diversity. Decreased inflammation markers IL-6 and serum LBP. Increased intestinal tight junctions. Improved cognitive function.	Clinical study
FMT (Sharma A et al., 2022) [169]	Patients with alcohol-associated acute-on-chronic liver failure	30 g of fresh stool homogenized with a hundred milliliters. Through a nasojejunal tube.	Improved short-term (28-day) and medium-term (90-day) survival rates. Enhanced resolution of clinical complications like hepatic encephalopathy and ascites; improved inflammation markers (IL-1β) and liver function.	Clinical trial

CFU, colony-forming unit; PFU, plaque-forming unit; *E. faecalis*, *Enterococcus faecalis*; LPS, lipopolysaccharides; ALT, alanine aminotransferase; AST, aspartate aminotransferase; ALD, alcoholic-related liver disease; AH, alcoholic hepatitis; LBP, lipopolysaccharide-binding protein; GGT, gamma-glutamyl transferase; LDH, lactate dehydrogenase; LDLC, low-density lipoprotein cholesterol; TG, triglycerides.

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
