# Peer review of "Gut Microbiota as Emerging Players in the Development of Alcohol-Related Liver Disease"

_biomedicines, 2024, doi:10.3390/biomedicines13010074_

Round 1

Reviewer 1 Report

Comments and Suggestions for Authors

The present review explored the impact of gut microbiota on the occurrence and development of alcohol related liver disease, discussed the potential mechanism of gut microbiome promoting the development of alcohol-related liver disease, and provided an effective early intervention method for alcohol-related liver disease. I have some comments to improve it:

1. Abstract section: remove the lines about figure 1 and add some lines about the main findings of your investigation.

2. The conclusion section need some lines about the future perspectives.

3. I suggest to draw a table to summarize the main findings of the potential therapeutic approaches.

4. The gut microbiota actively modulates the pro-inflammatory and anti-inflammatory cytokines, as well elucidated by recent literature data (doi: 10.3390/diseases12040069 - doi: 10.1089/jir.2019.0011). Add some lines about this topic.

5. The Mediterranean diet its a dietary regimen that suggest the moderate use of red wine, composed by different polyphenols (such as resveratrol). I suggest to add some lines about this topic (doi: 10.1080/10408398.2012.747484).

6. Add the aim of the review in the Introduction section.

Author Response

Comments 1: Abstract section: remove the lines about figure 1 and add some lines about the main findings of your investigation.

Response 1: Thank you for your valuable suggestions! We have moved Figure 1 to introduction and added some lines about the main findings of my investigation, which are located in lines 30-36, the red marked part.

Comments 2: The conclusion section need some lines about the future perspectives.

Response 2: Thank you for your valuable suggestions! We have added a few lines about future prospects in the conclusion, specifically lines 642-657, marked in red.

Comments 3: I suggest to draw a table to summarize the main findings of the potential therapeutic approaches.

Response 3: Thank you for your valuable suggestions! We have summarized the treatment approaches in two tables. One is the clinical (Table 2) and the other is the preclinical (Table s1). The two tables are detailed in the supplementary word document.

Comments 4: The gut microbiota actively modulates the pro-inflammatory and anti-inflammatory cytokines, as well elucidated by recent literature data (doi: 10.3390/diseases12040069IF: 2.9 Q2 IF: 2.9 Q2 - doi: 10.1089/jir.2019.0011IF: 1.9 Q4 ). Add some lines about this topic

Response 4: Thank you for your valuable suggestions! We fully agree with your suggestion. We added some lines in the intestinal microbiota and immunity section, specifically in lines 277-286, marked in red.

Comments 5: The Mediterranean diet its a dietary regimen that suggest the moderate use of red wine, composed by different polyphenols (such as resveratrol). I suggest to add some lines about this topic (doi: 10.1080/10408398.2012.747484IF: 7.3 Q1 ).

Response 5: Thank you for your valuable suggestions! I have added some talks about red wine to the treatment section, in lines 533-541, marked in red.

Comments 6. Add the aim of the review in the Introduction section.

Response 6: Thank you for your valuable comments! We have added the aim of the review in the Introduction section, in lines 68-73, marked in red.

Reviewer 2 Report

Comments and Suggestions for Authors

Recommendations:

Major issues:

1. Too many references, for a short narrative review.

2. Also to high iThenticate, reduce it.

3. Not interested in animal studies, this should address only clinical studies.

4. A lot of self-citation instances for some authors.

Minor issues:

1. Add narrative review in the title.

2.Talk about the microbiome in alcohol related pathologies like chronic pancreatitis. see this: https://doi.org/10.3390/diagnostics14090861 

Author Response

Major issues:

Comments 1: Too many references, for a short narrative review.

Response1: Thank you for your valuable comments! We streamlined the article's citations and removed 30 citations.

Comments 2: Also too high iThenticate, reduce it.

Response 2: Thank you for your suggestions! We have made some language-level modifications and strengthened the expressions in some parts of the article to reduce repetition.

Comments 3: Not interested in animal studies, this should address only clinical studies.

Response 3: Thank you for your suggestions! In the part about changes in intestinal microbiota, we have deleted the animal research content (in lines 153-160 and partly table 1,marked in red) and only left the clinical research. However, regarding the treatment part, we covered some preclinical studies. The first reason is that the treatment methods are constantly updated, and preclinical research is an important part and basis for conducting clinical research. The second reason is that this part of preclinical research can provide new insights into potential treatments. This review aims to explore the treatment of ALD by intestinal microbiota, including possible potential treatments (the mechanisms revealed by preclinical studies), so the treatment part includes clinical studies and preclinical studies.

Comments 4: A lot of self-citation instances for some authors.

Response4: Thank you for your suggestion! We carefully checked the number of self-cited articles and found that Professor Chu had 2 self-cited articles and Professor Yang had 1 self-cited article. We believe that these research results play an important supporting role in the conclusions of this article.

Minor issues:

Comments 1:Add narrative review in the title.

Response1:Thank you for your suggestions! Sorry, we checked the title again to make sure it fits the topic of our article.

Comments 2: Talk about the microbiome in alcohol related pathologies like chronic pancreatitis. see this: https://doi.org/10.3390/diagnostics14090861  

Response2:Thank you for your suggestions!We have added the relevant content. The specific location is in lines 161-169, marked in red.

Round 2

Reviewer 2 Report

Comments and Suggestions for Authors

Congratulations to the authors!